# How New Technologies Will Transform Total Knee Arthroplasty from a Singular Surgical Procedure to a Holistic Standardized Process

**DOI:** 10.3390/jcm14093102

**Published:** 2025-04-30

**Authors:** Elliot Sappey-Marinier, Thais Dutra Vieira, Axel Schmidt, Tarik Aït Si Selmi, Michel Bonnin

**Affiliations:** Centre Orthopédique Santy, Hôpital Privé Jean Mermoz, Ramsay Santé, 69008 Lyon, France; thaisdutravieira@hotmail.com (T.D.V.); axel.schmidt0310@gmail.com (A.S.); tarik.aitsiselmi@gmail.com (T.A.S.S.); bonnin.michel@gmail.com (M.B.)

**Keywords:** total knee arthroplasty, custom implants, artificial intelligence, robotic-assisted surgery, personalized alignment

## Abstract

Many new technologies focused mainly on improving surgical accuracy were first developed in total knee arthroplasty and have not yet shown significant value. These non-significant clinical improvements could potentially be explained by an inadequate target. In this current concept paper, the authors will describe how artificial intelligence (AI), robotic-assisted surgery, and custom implants allow the definition of new targets and the standardization of the TKA process. As paradoxical as it may be, new technologies in TKA will allow for better standardization in the overall way in which this procedure is carried out. Achieving this goal can be accomplished by incorporating AI-driven tools into the medical field. These tools are intended to enhance decision making, refine surgical planning, and increase the precision and consistency of surgical procedures. Moreover, custom implants with personalized alignment, beyond restoring native anatomy, will define new targets and standardize the whole TKA process.

## 1. Introduction

Total knee arthroplasty (TKA) is considered the gold standard for end-stage knee osteoarthritis [1]. TKA leads to good results in overall activity, excellent survivorship, and good functional improvement. Patient satisfaction remains one of the main aims after knee arthroplasty and defines a successful operation. However, despite improvements in prosthetic design and postoperative management, up to 20% of patients are dissatisfied with their prosthesis [2].

In the past, every step in the TKA process, from diagnosis to follow-up, was based on the surgeon’s personal judgment. Over the last decade, massive evolutions have occurred, leading to a control and scrutinization of the surgeon and more transparency in the process on a global scale (national registries, open medical literature, social media, etc.). Indeed, TKA has become more and more popular. Thus, there is a strong need for a standardization of the process to improve precision, repeatability, accuracy, and safety.

A recent review [3] highlighted new technologies developed to improve surgical accuracy and potentially improve patients’ clinical outcomes and satisfaction. First, patient-specific instrumentations have shown controversial results in improving limb alignment and implant positioning, and no differences have been found in clinical outcomes [4,5,6,7]. Second, sensors have been created with the aim to objectively measure ligament balance during TKA. TKA is considered balanced when the loading difference is inferior to 15 pounds between the medial and lateral compartments [8]. However, whether this definition applies to the native knee remains controversial. Third, accelerometer-based navigation has allowed improvements regarding accuracy in the intended alignment for implant positioning [9]. The benefits of the above remain controversial [10]. Thus, these new technologies have mainly been focused on improving surgical accuracy and have not yet shown significant value.

Those non-significant clinical improvements could potentially be explained by an inadequate target. To further improve outcomes after TKA, new technologies should better define the appropriate target for each patient benefiting from a TKA. They will allow one to consider TKA as an entire process, encompassing indication of TKA, preoperative planning, supply chain, workflow in the hospital, education, operative room instrumentation, implants, surgical technique, staff workload, traceability, and postoperative rehabilitation and assessment.

This article aims to present a standardized approach to total knee arthroplasty (TKA) through the integration of advanced technologies, including artificial intelligence, robotic-assisted surgery, and custom implants. By leveraging AI to develop personalized alignment strategies, incorporating robotics to enhance surgical precision, and utilizing custom-designed implants to restore the knee’s native anatomy, these innovations collectively contribute to improving the accuracy, consistency, and outcomes of TKA procedures.

## 2. Artificial Intelligence

Artificial intelligence (AI) can be defined as human intelligence exhibited by machines. It involves machine learning (ML) algorithms with reasoning ability and cognitive function execution [11].

ML, a subset of artificial intelligence (AI), utilizes diverse algorithms and computational frameworks to identify patterns and improve outcomes based on historical data [12]. Deep learning (DL) is a subdivision of ML that uses large volumes of previous training information to be able to perform complex tasks [13]. DL operates from an artificial neural network (ANN) composed of neurons hierarchically organized. A neural network processes information in sequential layers, beginning with basic input analysis. Each layer refines the data further, drawing on outputs from preceding layers, and passes them forward for more complex interpretation. This hierarchical processing continues iteratively until the network can perform the desired task—such as analyzing, classifying, or segmenting medical images like radiographs or CT scans [14].

Convolutional neural networks (CNNs) are a specialized and powerful type of deep learning architecture particularly suited for image analysis. They process visual data through multiple layers composed of filters that learn to recognize features progressively. Unlike traditional deep learning models, CNNs leverage spatial relationships between pixels, allowing for more efficient computation and reduced parameter complexity at each layer [15]. An important strength of both deep learning and CNNs lies in their capability to autonomously identify relevant features in input data once labeled outputs are available for training. Since CNNs rely on repeated exposure to data, their accuracy improves with larger datasets. Additionally, deep learning models typically offer faster data processing and require less computational effort compared to many other AI methodologies [16].

Various aspects of AI, including ML and DL, are currently being applied across the preoperative, intraoperative, and postoperative stages of TKA (Figure 1).

ML can help predict costs before primary TKA and length of stay (LOS) [18], 30-day readmission [19], postoperative dissatisfaction after TKA [20], transfusion risk after primary TKA [21], TKA component size [22], and poorest outcomes [23] (any extension of the femoral component or excessive femoral flexion and deviation from native tibial slope predict poorer outcomes). ML and predictive models can help estimate postoperative outcomes and patient satisfaction and thus gain an understanding of future patient and surgeon decision making [24,25]. These models analyze the impact that variables such as patient-specific demographic data, functional scores and preoperative pain [26], comorbidities [27], psychological characteristics [28,29,30], socioeconomic indicators [28], and/or perioperative recovery location can have on the patients’ clinical outcomes. Predictive models that incorporate large-scale datasets—often referred to as big data—combined with objective preoperative 3D anatomical evaluations tend to yield greater accuracy compared to models built on smaller, more limited data samples [31]. To date, existing predictive models have not yet reached the level of diagnostic intuition and decision-making expertise demonstrated by experienced surgeons [32] nor have they become fully practical for routine clinical application. The validation of these predictive models is currently in the pre-clinical stage [33]. Postoperatively, remote patient monitoring systems can also add more information, allowing for a more comprehensive assessment of patients undergoing TKA in terms of mobility and rehabilitation compliance [34].

The DL system can precisely identify the presence of a TKA, differentiate specific arthroplasty patterns, and also recognize and clinically classify knee osteoarthritis as accurately as an orthopedic surgeon [35,36]. DL can detect the loosening of prostheses from X-rays [37]. ANNs can predict length of stay and hospital charge and discharge arrangements before primary TKA [38]. CNNs can discriminate different types of implants with nearly perfect fidelity and also detect implant loosening signs from radiographs [39]. Moreover, a recent study highlighted the benefit of using CNNs for a standardized interpretation of postoperative knee X-rays and indicated the potential for their use in clinical practice [40].

To improve surgical training, immersive virtual reality (VR) platforms have been developed and are increasingly incorporating elements of artificial intelligence (AI). The integration of AI—such as real-time performance assessment, adaptive feedback, and predictive modeling—enhances VR’s educational value. In this context, immersive VR combined with AI algorithms can be regarded as an AI-enhanced learning tool. These systems enable users to assess decisions related to implant selection and positioning, monitor efficiency, detect technical errors, and interact effectively with surgical instruments like fluoroscopes and retractors. By acquiring technical and cognitive practice over time, surgeons can improve their dexterity, resulting in lower implant mis-alignment rate and surgical complications in primary and revision TKA [41].

Thus, AI technology helps orthopedic surgeons provide patient-specific care around TKA in terms of preoperative health optimization, clinical management, decision support, strategic resource planning, early intervention, surgical planning, and training and education. However, the clinical effectiveness and safety of AI-based knee arthroplasty remain to be validated.

## 3. Robotic-Assisted Surgery

We believe that one major new technology in orthopedics is robotic-assisted surgery, which is part of the AI realm. It has been widely adopted in recent years, allowing reproducible and accurate bone preparation through the robotic interface. The main objective of robotic surgery is not to substitute the surgeon but enhance their performance. The robotic interface allows an assessment of ligament balancing before any bone cuts and adapts the implant’s positioning according to this assessment. Different tools exist to perform the ligament assessment as objectively as possible, such as mechanical tools like retractors or spoons. However, many surgeons still perform ligament assessments by providing manual varus and valgus stress.

Robotic-assisted knee arthroplasty involves a machine that can perform intricate tasks automatically, particularly those programmed by a computer. This technology integrates data from preoperative imaging or real-time intraoperative surface mapping, along with key bone landmarks such as tibial and femoral alignment, to ensure optimal ligament balance. Robotic systems can be categorized into three types: passive, semi-autonomous, and autonomous (Figure 2).

A passive system generates a 3D virtual model for precise preoperative planning but does not perform bone preparation. Autonomous and semi-autonomous systems have built-in safety measures to prevent excessive bone removal outside the designated 3D plane. The semi-autonomous robotic-assisted system represents a prime example of an artificial intelligence (AI)-driven tool that combines the advantages of a navigation system and an autonomous robotic system. This system gathers data such as algorithms for aligning bones and implants, as well as assessments of soft-tissue balance, to suggest surgical plans. These plans can then be fine-tuned based on the surgeon’s specific preferences and objectives. A robotic arm enables the performance of bone resections or the precise placement of a cutting guide, all while providing real-time automatic feedback that adjusts to knee movements and the progress of the cuts being made. Over time, the algorithms within robotic systems incorporate machine learning models to enhance surgical planning based on prior surgeries. During the surgical procedure, a feedback loop is established to monitor and control bone cutting or the positioning of the cutting guide. This level of control enhances the surgeon’s precision and reduces the likelihood of errors. It is important to note that robotic systems are not designed to replace surgeons but rather to serve as highly accurate and consistent tools to assist them. The primary advantage of these robotic systems lies in their ability to achieve precise and reproducible bone preparation through the robotic interface, regardless of the specific system being utilized [42,43]. The majority of presently accessible robotic platforms evaluates ligament balance during surgery by considering factors such as bone cutting and implant placement. The benefits of robotic-assisted TKA include achieving precise alignment of the knee, accurate positioning of the implants, maintaining proper ligament balance, and safeguarding the integrity of soft tissues [44,45,46,47]. In most controlled studies, it has been indicated that there are improved functional outcomes in the short term compared to conventional TKA, but these advantages still remain to be proven considering mid- to long-term results [48,49,50,51,52]. Robotic-assisted total TKA comes with some drawbacks. These include substantial initial costs for acquiring the equipment and necessary supplies for the operating room. Additionally, there is a need for extensive training for both the surgeon and the surgical team to ensure the safe and efficient use of robotics. The system involves specialized hardware with bone-tracking devices and a bulky robotic unit, which can be inconvenient. The surgical procedures often take longer due to the learning curve associated with robotic assistance, which can compromise cost-effectiveness. Furthermore, robotic systems are typically designed to work with specific implant types, limiting their versatility. In the early stages of surgery, the assessment of joint laxity is performed manually, leading to potential inaccuracies and a lack of consistency.

## 4. Custom Implant and Personalized Alignment

Surgeons now have the option to select from a variety of component sizes, including asymmetrical tibial components in some cases, as well as femoral components with narrow or standard widths. However, over the past decade, many studies have shown a large variability in the human anatomy among patients, including dimensions, width, trapezoidicity, and trochlea variations, all statistically independent. Hence, the “off-the-shelf” TKA implants available have limitations in terms of accommodating the diverse size and shape variations found in human knees. Studies have shown that oversizing can occur in as many as 76% of cases in relation to the femur and up to 90% in relation to the tibia [53,54].

Customized implants are designed with the goal of replicating the native, pre-arthritic knee anatomy and its biomechanics. The primary objectives are as follows:Ensuring an optimal fit between the bone and implant to prevent either overhang or inadequate coverage;Enhancing ligament balance by avoiding laxity caused by asymmetric resection;Improving stability and joint movement in the middle range of flexion by restoring the knee’s native curvatures;Enhancing the tracking of the patella in relation to the femur by replicating natural femoral torsion and creating a customized trochlea;Facilitating the restoration of the limb’s original, pre-arthritic alignment.

Over this last decade, a paradigm shift has occurred in restoring native alignment via a personalized alignment instead of a systematic mechanical alignment. Several personalized alignments were described, including kinematic alignment, restricted kinematic alignment, inverse kinematic alignment, and functional alignment [55]. For instance, a personalized algorithm was described with the Origin^®^ prosthesis. Certainly, the Origin Alignment© maintains its adherence to safe parameters, particularly regarding tribology, fixation, joint line obliquity (JLO) within a range of ±5°, and postoperative alignment within a range of ±3°. Given these considerations, it is estimated that around 75% of osteoarthritic knees can be effectively treated using the Origin Alignment©.

Consequently, we argue that restoring the knee’s native anatomy, including proper limb alignment, may enhance functional outcomes in TKA. Moreover, customization of the bone cuts and implants to each patient’s unique needs enables engineers to minimize implant thickness, weight, and the extent of bone resections required to a great extent.

Cooperation is required between the engineers and the surgeon to design and manufacture a custom TKA, and this process takes 6–8 weeks. A 3D model, obtained by transforming a series of 2D scanned images of the knee joint, is used to manufacture specific instrumentation and a customized implant (Figure 3) using additive manufacturing/3D printing technologies. The surgeon validates the implant design and operative planning. Personalized cutting blocks are used to perform tibial and femoral cuts, with the hope of using robotics in the near future. All surgical parameters, such as limb alignment, implant positioning, and size, are decided preoperatively.

The short-term results of custom TKA are very promising. Indeed, performing a custom TKA following the Origin^®^ prosthesis principles facilitated restoration of constitutional coronal alignment [56] and granted very good clinical outcomes [57,58] even in knees that had prior osteotomies or extra-articular fracture sequelae [59].

## 5. Discussion

In this paper, we showed that TKA is not just an implanting process but a holistic one. We hold a firm conviction that TKA requires enhancements in three key areas:(i)The development of a personalized alignment strategy, facilitated by artificial intelligence (AI) [55,57];(ii)The enhancement of surgical accuracy through the integration of cutting-edge technologies such as robotics [3,42,46];(iii)The restoration of the knee’s native anatomy by implementing custom-designed implants [56,57,58].

Artificial intelligence will help improve patient selection, thanks to predictive models which are still imprecise [18,20,24]. However, many other tools are already available and very helpful throughout the entire process of a TKA procedure, from preoperative assessment to postoperative status, including patient rehabilitation [34,37]. This is just the beginning of the AI era. Many other solutions will appear in the next few years while connecting all data gathered in the health care system (Figure 4) [25,36].

It is important to highlight that customized cutting guides and navigation systems, when employed with standard implants, have not shown substantial advantages in terms of outcomes and patient contentment [4,5,6,7]. Likewise, while robotic surgery has proven to be more dependable and accurate, it may not address the primary challenges stemming from the non-anatomic design of the implant itself [3,42,44,47]. The next generation of robotics should be an image-based, tracker-less, surgeon-friendly robot with an accurate and reproducible ligament balance tool [8,43,45].

Customized TKA offers numerous benefits for surgeons. Firstly, it simplifies the surgical process by automatically addressing several surgical challenges through the conservation or restoration of the patient’s native knee anatomy [56,57]. This includes (i) adjustments to femoral and tibial rotation during the design phase, (ii) easier balancing, particularly in mid-flexion, due to maintaining natural curvatures and joint alignment, and (iii) eliminating the need for size adjustments as the bone–implant fit is optimized [53,54]. Secondly, preoperative planning for alignment and implant positioning enhances surgical precision and provides a safety net for the surgeon [55,56]. Thirdly, this technology can be especially helpful in challenging cases, such as (i) patients with post-traumatic extra-articular deformities, (ii) hardware positioned close to the joint surface, where the instruments and the implant are designed to avoid conflict, (iii) previously operated or infected bones, (iiv) and bones with unique/extreme anatomical characteristics that make standard TKA implantation difficult [57,59].

Furthermore, this technology streamlines hospital management by simplifying the surgical workflow. A single package containing customized implants and instruments tailored to the patient’s needs eliminates the need for large inventories of implants and multiple instrument trays. A recent cooperative study indicated improved operational efficiency and team well-being in custom TKA compared to conventional methods [57]. Finally, this technology significantly reduces the demand for sterilization, leading to substantial cost savings and positive ecological consequences by reducing water consumption for sterilization purposes.

Therefore, we believe that these three pillars will improve TKA results by standardizing the entire process, including patient selection, preoperative planning, supply chain, workflow in the hospital, surgeon and patient education, instrumentation, restoring anatomy, surgical technique, traceability, and postoperative evaluation.

## 6. Conclusions

As paradoxical as it may be, new technologies for TKA will be the ones allowing for the better standardization of the overall process involved in this procedure. This will be possible by including AI-based tools to enhance decision making, surgical planning, and the accuracy and consistency of the procedures. Moreover, custom implants with personalized alignment, beyond restoring native anatomy, will define new targets and standardize the whole TKA process.

## Figures and Tables

**Figure 1 jcm-14-03102-f001:**
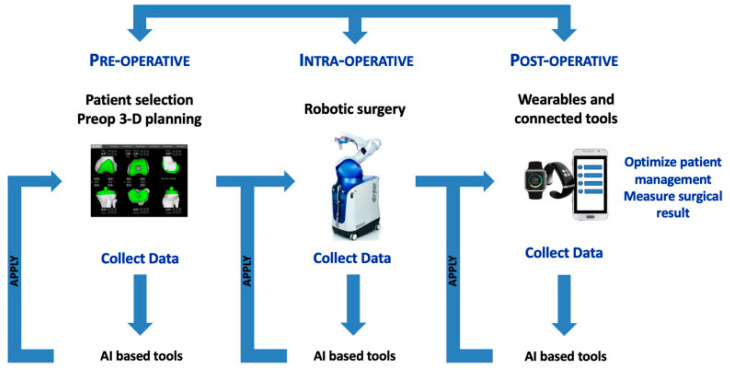
Diagram illustrating the feedback loop concept, where data collected before, during, and after surgery through connected tools are integrated to generate large-scale insights (“megadata”) that inform and refine the surgical plan (reprinted with permission) [17].

**Figure 2 jcm-14-03102-f002:**
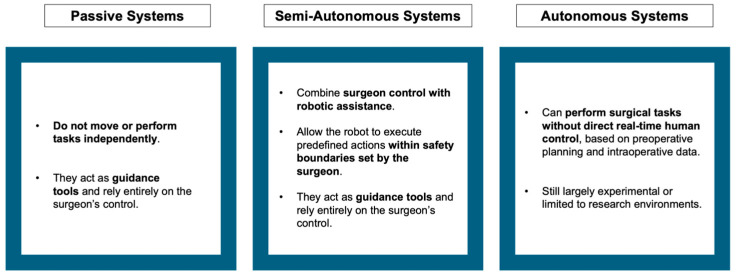
Classification of robotic systems used in total knee arthroplasty (TKA). The diagram illustrates three categories: passive systems, which provide visual or mechanical guidance without executing surgical tasks; semi-autonomous systems, which assist the surgeon by performing specific actions such as bone cutting within predefined limits; and autonomous systems, capable of executing surgical steps independently based on preoperative planning. The level of robotic involvement increases progressively from left to right.

**Figure 3 jcm-14-03102-f003:**
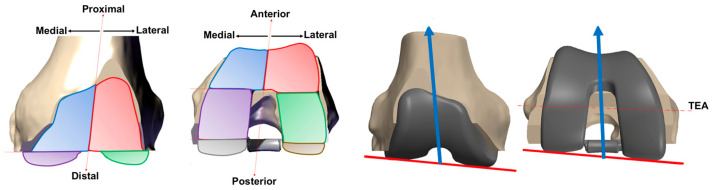
Customization possibilities of all independent areas allowing one to design the trochlear shape independently from the joint line’s orientation and the trans epicondylar axis (TEA).

**Figure 4 jcm-14-03102-f004:**
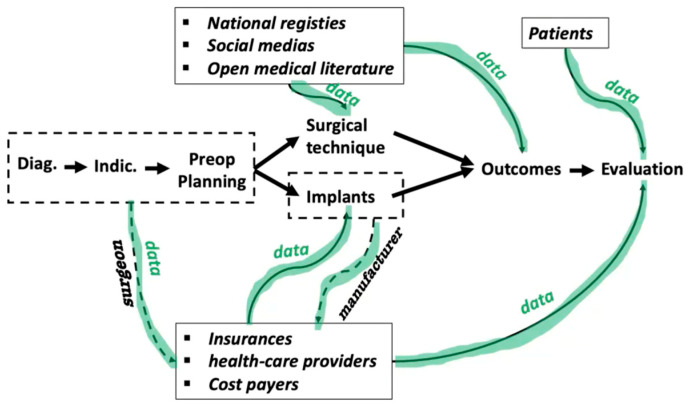
Artificial intelligence will connect all data gathered to improve health care in general.

## Data Availability

Not applicable.

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
