# Peer review of "How New Technologies Will Transform Total Knee Arthroplasty from a Singular Surgical Procedure to a Holistic Standardized Process"

_jcm, 2025, doi:10.3390/jcm14093102_

Round 1

Reviewer 1 Report

Comments and Suggestions for Authors

Despite a well-performed surgery of total knee arthroplasty (TKA), some patients, unfortunately, have less favorable outcomes characterized by multiple different issues such as ongoing pain, swelling, instability, infection, inflammation, risk of bone fracture and arthrofibrosis. Those complications need more research effort and models to address. The authors of this manuscript have broadly reviewed the recent development and application of artificial intelligence (AI) in total knee TKA.

Importantly, authors wave a flag that application of AI, modern technology, will transform TKA from a singular surgical procedure to a global standardized process. Importantly, authors present the advantages of application of AI technology in TKA, such as precisely classify the knee osteoarthritis, apply robotic assisted surgery procedures, design personalized implant, and detect loosening of protheses from X-ray. These are important for further study toward clinical application of AI technology.

The authors make significant contribution to the future development of orthopaedic surgery and research. The manuscript is an elevated level of perspective paper.

Minor comments and suggestion:

Line 228: Labels in Figure 1 are not readable.

Line 232: Authors wrote, “we showed that TKA is not just an implant but a global process”. Suggestion: Use word “implanting.”

Line 240: Authors stated that “Artificial intelligence will help in improving patient selection thanks to predictive models which are still imprecise.” A comma is missing between words selection and thanks.

Author Response

Dear Reviewers,

We would like to sincerely thank you for your thoughtful and constructive feedback on our manuscript. Your comments have been incredibly valuable in improving the quality and clarity of our work. We have carefully addressed each of your suggestions and provided our responses below.

Elliot Sappey-Marinier

Minor comments and suggestion: 

  • Line 228: Labels in Figure 1 are not readable.

This was corrected.

  • Line 232: Authors wrote, “we showed that TKA is not just an implant but a global process”. Suggestion: Use word “implanting.”

This was updated in line 232: In this paper, we showed that TKA is not just an implanting but a global process. We hold a firm conviction that TKA requires enhancements in three key areas

  • Line 240: Authors stated that “Artificial intelligence will help in improving patient selection thanks to predictive models which are still imprecise.” A comma is missing between words selection and thanks.

This was updated in line 240: Artificial intelligence will help in improving patient selection, thanks to predictive models which are still imprecise.

  • Lines 66–68: Consider expanding this section with a more detailed explanation, especially in relation to the aim stated in lines 234-239.

This was updated in lines 66-72: This article aims to present a standardized approach to total knee arthroplasty (TKA) through the integration of advanced technologies, including artificial intelligence, robotic-assisted surgery, and custom implants. By leveraging AI to develop personalized alignment strategies, incorporating robotics to enhance surgical precision, and utilizing custom-designed implants to restore the knee’s native anatomy, these innovations collectively contribute to improving the accuracy, consistency, and outcomes of TKA procedures.

  • Line 121: Please clarify why immersive virtual reality is considered an AI-based learning tool. What specific AI elements are involved?

This was updated in lines 124-133:

To improve surgical training, immersive virtual reality (VR) platforms have been developed and are increasingly incorporating elements of artificial intelligence (AI). The integration of AI—such as real-time performance assessment, adaptive feedback, and predictive modeling—enhances VR educational value. In this context, immersive VR combined with AI algorithms can be regarded as an AI-enhanced learning tool. These systems enable users to assess decisions related to implant selection and positioning, monitor efficiency, detect technical errors, and interact effectively with surgical instruments like fluoroscopes and retractors By acquiring technical and cognitive practice over time, surgeons can improve their dexterity resulting in lower implant mis-alignment rate and surgical complications in primary and revision TKA.

Lines 232–278: Please provide references for the discussion to support your statements.

This has been updated throughout the manuscript.

  • Additionally, I suggest including illustrative figures for all the technologies discussed (not just for the personalized implant) to enhance readability and comprehension.

Figure 1 and 2 were added for Artificial Intelligence and Robotic-assisted surgery.

  • Furthermore, I suggest reviewing the high number of self-citations.

Thank you for your observation. In response, we reviewed all self-citations and removed four of our own references, retaining only those directly relevant and essential to the manuscript's context.

Reviewer 2 Report

Comments and Suggestions for Authors

The aim of this article is to analyze the approach to the Total Knee Arthroplasty (TKA) process through the integration of new technologies such as artificial intelligence (AI), robotic-assisted surgery, and custom implants.

The authors provide a general overview of TKA treatment, highlighting the advantages and limitations of these emerging technologies. Despite advancements, clinical outcomes for patients remain suboptimal, raising questions about the real impact of such innovations. While TKA procedures have become increasingly standardized, the application of customization through implants and technologies remains controversial. While these approaches may enhance surgical accuracy, they often lack substantial clinical benefits. This gap is due to the lack of an approach that considers the entire process, from individual patient needs and preoperative planning to hospital logistics and resource availability.

The article offers a review of the technologies analyzed, explaining their basic mechanisms and their specific roles within the context of TKA. Artificial intelligence contributes to multiple phases of the TKA workflow, offering improvements not only from a medical point of view but also from a socioeconomic perspective. AI can assist in imaging analysis, training, and the delivery of personalized patient care. Robotic-assisted surgery is evolving toward full automation, supported by AI systems that learn progressively from each procedure. Lastly, the use of custom implants and personalized knee alignment enables replication of the patient’s unique anatomy and biomechanics.

  • Lines 66–68: Consider expanding this section with a more detailed explanation, especially in relation to the aim stated in lines 234-239.
  • Line 121: Please clarify why immersive virtual reality is considered an AI-based learning tool. What specific AI elements are involved?
  • Lines 232–278: Please provide references for the discussion to support your statements.
  • Additionally, I suggest including illustrative figures for all the technologies discussed (not just for the personalized implant) to enhance readability and comprehension.
  • Furthermore, I suggest reviewing the high number of self-citations.

Author Response

(The authors gave the same response as above.)
